# Emerging Role of Deuterium/Protium Disbalance in Cell Cycle and Apoptosis

**DOI:** 10.3390/ijms24043107

**Published:** 2023-02-04

**Authors:** Nataliya V. Yaglova, Ekaterina P. Timokhina, Sergey S. Obernikhin, Valentin V. Yaglov

**Affiliations:** Laboratory of Endocrine System Development, A.P. Avtsyn Research Institute of Human Morphology of Federal State Budgetary Scientific Institution, Petrovsky National Research Centre of Surgery, 119991 Moscow, Russia

**Keywords:** deuterium, deuterium-depleted water, deuterium-enriched water, protium, D/H gradient, proliferation, apoptosis, cell cycle, tumor

## Abstract

Deuterium, a stable isotope of hydrogen, is a component of water and organic compounds. It is the second most abundant element in the human body after sodium. Although the concentration of deuterium in an organism is much lower than that of protium, a wide variety of morphological, biochemical, and physiological changes are known to occur in deuterium-treated cells, including changes in fundamental processes such as cell division or energy metabolism. The mode and degree of changes in cells and tissues, both with an increase and a decrease in the concentration of deuterium, depends primarily on the time of exposure, as well as on the concentration. The reviewed data show that plant and animal cells are sensitive to deuterium content. Any shifts in the D/H balance outside or inside cells promote immediate responses. The review summarizes reported data on the proliferation and apoptosis of normal and neoplastic cells in different modes of deuteration and deuterium depletion in vivo and in vitro. The authors propose their own concept of the effects of changes in deuterium content in the body on cell proliferation and death. The altered rate of proliferation and apoptosis indicate a pivotal role of the hydrogen isotope content in living organisms and suggest the presence of a D/H sensor, which is yet to be detected.

## 1. Introduction

Chemical elements are characterized by their atomic number, that is, the number of protons in the nuclei of its atoms. In addition, the atoms of some elements can contain a different number of neutrons in the nucleus and are therefore called isotopes. Stable isotopes, unlike radioisotopes, do not have the ability to spontaneously decay. About 2/3 of the elements have more than one stable isotope, the number of which varies from two to ten [1]. Isotopes of hydrogen, carbon, oxygen, and nitrogen are abundant in the nature of living organisms, while other stable isotopes of biogenic elements, such as selenium, magnesium, and sulfur are also of great importance. There are usually more light isotopes than heavy ones. For example, the proportion of carbon which is C-12 in living organisms is 98.89%, and C-13 is 1.11%; nitrogen N-14 is 99.63%, and N-15 is 0.37%; the proportion of oxygen O-16, O-17, and O-18 is 99.759, 0.037, and 0.204%, respectively; and finally, the proportion of hydrogen isotope H-1 is 99.985%, and H-2 (D) is 0.015% [2].

Isotopes of hydrogen and oxygen are of particular importance among all non-radioactive substances, as they are components of both organic and inorganic compounds as well as water, an indispensable solvent for all biological substances, where the vast majority of biochemical reactions take place. In this regard, the isotopic composition of water is of great importance in the implementation of physiological processes.

The relevance of deuterium concentrations is difficult to overestimate. Deuterium is a component of water but also of numerous organic compounds. It is the second most abundant element in the human body after sodium. Its concentration in blood plasma is four times more than potassium, six times more than calcium, ten times more than magnesium, and much more than the content of essential macro- and microelements, such as fluorine, iron, iodine, copper, manganese, and cobalt. It is important to note that the deuterium content in the blood plasma of healthy animals is 3–6 ppm higher than the level of deuterium in drinking water [3,4,5]. Although the concentration of deuterium in an organism is much lower than that of protium and reflects the D/H ratio in natural nonorganic and organic compounds, a wide variety of morphological and physiological changes have been shown in cells enriched with deuterium, including changes in fundamental processes such as cell division or energy metabolism [6,7].

## 2. Deuterium Content in Natural Waters

The water of cells and tissues is constantly enriched with deuterium. Two main sources of deuterium supply have been found: an outer source and an inner source. The first is ingested natural water, and the latter is intrinsic water that is produced as result of the transformation of metabolites, including the reduced form of nicotinamide adenine dinucleotide (NAD) in the respiratory chain of mitochondria [8,9].

According to the International Standard Marine Ordinary Water (SMOW), the deuterium content in drinking water is 155.76 ppm. The average range of deuterium concentrations in drinking water was determined after evaluation of the isotopic composition of bottled and packaged samples of water from around the world by scientific groups from different countries. Its value is approximately 130 to 160 ppm and depends on the geographical region [10,11,12]. It should be noted that fluctuations in the isotopic composition of water in the human organism can have a wider range due to the consumption of both surface water and water from other sources (artesian, glacial, mineral springs) and dietary preferences, such as the predominant intake of lipid-based nutrients. Lipid-based nutrition have demonstrated about a two-fold increase the production of intracellular water compared to carbohydrate nutrition, mainly due to cooperative mitochondrial and peroxisomal lipid–substrate oxidation to H_2_O [13,14]. According to scientific reports the concentration of hydrogen isotopes in oceanic water has changed by 25 ± 5% since the Archean era. This fact allows a hypothesis to be put forward that during the active evolutionary development of living organisms, the content of deuterium in natural water was lower than in modern times, and in this regard, “light” water may be considered a stimulus for the activation of certain processes in the organism [15].

## 3. Identified Molecular and Physiological Effects of Deuterium

Chemical reactions involving different stable isotopes of the same atoms can have different rates. This phenomenon is known as the kinetic isotope effect. For most biogenic elements, this effect is very low [16]. The kinetic isotope effect of hydrogen isotopes is the strongest due to the large difference in atomic mass and nuclear spin between protium and deuterium. Thus, the substitution of protium by deuterium in chemical bonds significantly increases the bond breaking energy and affects the relative rate of chemical reactions. Kinetic isotope effects are classified into primary and secondary. More pronounced isotope effects are observed when a bond containing deuterium is localized directly at the site of the reaction. Such effects are called primary. Deuterium atoms located near but not in the reaction center produce secondary isotope effects that are usually much weaker than the primary ones [17].

A number of studies have examined the effects of reduced levels of deuterium on DNA. It has been reported that deuterium atoms getting into hydrogen bonds between pairs of the nitrogenous bases of the DNA molecule can affect the probabilities of the occurrence of open states (breaks in hydrogen bonds between bases) [18]. Consumption of deuterium-depleted water (DDW) significantly reduces the number of DNA single-strand breaks, which generally indicates an increase in the efficiency of cell defense systems. It is known that a DNA molecule in natural conditions has one deuterium atom for every 6400 hydrogen atoms, and a change in this ratio can lead to a change in the mutation frequency of the DNA molecule during the evolution of living systems [19]. Taken together these data suggest a higher efficacy of proliferation and a pool of investigation and support the idea to investigate the effects of DDW on cell proliferative activity and antitumor effects, as well as changes in the endogenous antioxidant system and non-specific resistance of the organism [20,21]. The results have shown that studies in this field are promising, as very interesting and encouraging data were obtained on the effects produced by shifts in protium and deuterium body content on cell proliferation and apoptosis [22,23,24,25]. However, the expansion of these studies has led to the emergence of diametrically opposed data, the analysis of which shows that the evaluation of certain processes at the cellular level requires fundamental information about the metabolism of hydrogen isotopes in the organism.

Some investigations have revealed significant effects of deuterium-enriched water (DEW) on physiological processes and biochemical reactions in the organism and showed that the degree of an effect depends directly on the concentration of deuterium in the consumed water, as well as on the period of exposure. The first experiments in the 1960s with DEW showed that the replacement of 25% of protium in tissues by deuterium leads to sterility in mammals, whereas higher concentrations result in a rapid lethality of the animals [26,27]. At the same time, the effects of short-term exposure to DEW on the organism were not studied, and there were no data on the effect of lower concentrations of deuterium in water on cells and tissues. The first studies in this area began only in the last decade, and at present, this topic is little studied, and the results are fragmentary. The studies have shown that DEW is less harmful to unicellular organisms. Various bacteria have lived in fully deuterated environments, for example Escherichia coli and Bacillus subtilis, as well as fungi, such as *Aspergillus niger* [27,28]. Cyanobacteria also proliferated actively at high concentrations of D_2_O. However, the growth of their colonies was slower and accompanied by various morphological abnormalities such as discoloration and increased cell size. Investigations into organisms capable of living and reproducing in a deuterated environment have identified some species that are able to adapt to heavy water. For example, the symbiotic organism medusomycete (tea fungus) was found to adapt to 98% D_2_O in just 24 h. Adaptation was monitored using the rate of glucose utilization and the accumulation of intermediate metabolites ethyl alcohol and acetic acid, measured using nuclear magnetic resonance. When transplanted into heavy water, the growth of the symbiotic organism was suspended for a day and then resumed, although at a lower rate than in the control [29]. The abovementioned data show that both DEW and DDW can induce some shifts in the functional activity of healthy organisms and suggest similar effects in pathology, which may become a tool in combating various functional disorders of the organism.

## 4. Physiological Effects of Altered D/H Gradient

It is known that the reduction/increase of deuterium levels during the consumption of DDW/DEW in organs and tissues occurs at different rates. Deuterium depletion or deuteration implicates the formation of an isotopic gradient and the intensification of D/H-exchange reactions, which result in a significant change in deuterium content, first in the blood plasma and then in the tissues of the liver, kidneys, and heart. It is known that the most active reactions of D/H isotope exchange occur in compounds that have atoms with unshared electron pairs and are capable of forming intermediate reaction complexes by the formation of hydrogen bonds with the synchronous transition of protons (H^+^) and deuterons (D^+^) from one molecule to another. Therefore, this exchange is observed more frequently in compounds that have hydroxyl (-OH), less frequently in thiol groups (-S-H) and primary and secondary amino groups (-NH2 and = N-H) but not in hydrocarbon bonds (R3C-H(D). This exchange is practically impossible under natural conditions, which partially explains the fact of incomplete isotopic D/H exchange in biological objects, where most hydrogen atoms are connected with carbon atoms [30]. The emergence of a D/H-gradient between the blood and the tissues of the body in the early stages of the deuterium level reduction leads to a strain on the body’s nonspecific defense system. This period of “pre-adaptation” lasts, on average, up to 14 days [31]. This phenomenon explains to some extent the stimulatory effect of DDW cells on the immune system, especially in conditions of induced inflammation. Bild et al. showed that 15-day consumption of DDW (30 ppm) resulted in a statistically significant increase in the inflammatory response in mice after subcutaneous injections of lipopolysaccharide pellets, demonstrated by higher percentages of neutrophils and lymphocytes in the blood and an increased phagocytic ability of the peripheral blood polymorphonuclear leukocyte [32]. A similar stimulating effect of the D/H-gradient was observed in studies of the thyroid gland. The short-term consumption of DDW for 24 h resulted in a marked stimulation of the thyroid gland functional activity. At the same time, the substitution of tap water with a deuterium content of 150 ppm for a 50% D_2_O for 24 h evoked similar changes in the functioning of the pituitary–thyroid axis, which included increased thyroxine production and decreased secretion of thyroid–stimulating hormones [33]. The similarity of changes during short-term exposure to DDW and DEW indicates that the abrupt change in the D/H gradient, both upward and downward, of the plasma deuterium concentration compared to the intracellular one, is accompanied by a stress effect on the organism and demonstrates a stimulating effect on the thyroid gland. It is worth noting that longer use of both DDW and DEW leads to suppression rather than activation of various processes in healthy cells and tissues. Thus, the consumption of DDW by rats at a concentration of 10 ppm for 14 days suppressed secretory processes and even evoked morphological changes in the thyroid parenchyma, as well as functional disorders of the hypothalamic–pituitary complex. The diminished synthesis of thyroid hormones was provoked by the lack of thyroid-stimulating hormones [34]. Mice treated with DEW at a concentration of 30,000 ppm for more than 7 days demonstrated a significant loss of thymus and spleen masses and increased cell death in the lymphoid organs. In addition, deuteration resulted in a dose-dependent suppression of specific antibody production after a subcutaneous injection of tetanus anatoxin [35]. These results demonstrate a similarity of changes in physiology after short-term bilateral shifts in the D/H gradient and deuterium content and the opposite of the effects evoked by prolonged exposure to deuteration and deuterium depletion of the organism.

## 5. Effect of D/H Shift on Cell Proliferation and Apoptosis

To date, a large number of papers have been published on the influence of the isotopic composition of consumed water on such pivotal processes as proliferation and apoptosis of both normal and tumor cells. For a better understanding, we provide basic information about the process of eukaryotic cell division.

### 5.1. Cell Proliferation and Its Control

The cell cycle is known as a period of cell life from one division to another or from division to death. It includes interphase (period of cell growth) and cell division (mitosis) itself. The interphase consists of three phases. After mitosis, the cell enters the presynthetic or G1 phase, then goes into the synthetic or S-period and then into the postsynthetic or G2 period. The G2 period ends the interphase, and after it, the cell enters into the next mitosis (Figure 1). The cell cycle has a number of checkpoints that play an important role in protecting the normal genome from damage [36,37,38]. Due to the regulation of the cell cycle, DNA replication and subsequent cell division is accurately controlled in normal cells, and the loss of genetic information is prevented [39]. Cell cycle has its own mechanisms of control called checkpoints. If a cell passes a checkpoint successfully, it enters the next phase of the cell cycle. If a checkpoint reveals some abnormalities, e.g., DNA damage or impaired chromatid attachment to the spindle microtubules, it prevents a cell from entering the next stage [40,41,42]. There are at least four checkpoints of the cell cycle: a checkpoint in G1, which checks for intact DNA before entering the S phase; a checkpoint in S phase, which checks for correct DNA replication; and a checkpoint in G2, which checks for damage missed during previous checkpoints or obtained during subsequent stages of the cell cycle. The G2 phase detects the completeness of DNA replication, and the cells in which the DNA is under-replicated do not enter mitosis. At the control point of the division spindle assembly, it is checked whether all kinetochores are attached to microtubules [43,44,45]. The irreversible damage of DNA activates proapoptotic molecules and induces the programmed cell death. The duration of the cell cycle varies between cell types. Rapidly multiplying cells of adult organisms, such as hematopoietic or basal cells of the epidermis and small intestine, may enter the cell cycle every 12–36 h. Short cell cycles (about 30 min) are observed in eggs of echinoderms, amphibians, and other animals. Under experimental conditions, many cell culture lines have cell cycles of about 20 h. The duration of the resting period between divisions in most actively dividing cells is about 10–24 h [46,47,48,49].

### 5.2. Implication of D/H Disbalance in Proliferation of Normal Cells

The first experiments with heavy water showed a marked delay in cell division in deuterated organisms. For example, 10% DEW has been shown to arrest the cell cycle of the egg cells of sea urchin (*Arbacia punctulate*) at all stages of mitosis and cytokinesis [50]. After returning to standard medium, a multiple, rapid, and heterogeneous acceleration of division was observed. Thus, metabolic processes and chromosome doubling were not irreversibly blocked. More recent experiments have shown that exposure to DEW increases the fraction of cells in S and G2/M phases, indicating that DNA replication, mitosis, and/or cell division are blocked. One of the mechanisms by which DEW blocks cell division is the stabilization of tubulin demonstrated in vitro, as well as the acceleration of tubulin polymerization by DEW, which is attributed to increased intra- and/or intermolecular hydrophobic interactions between tubulin molecules [51,52]. The effect of deuterium content in the culture medium on the proliferation rate of cultured stem cells derived from human adipose tissue has been shown in in vitro studies. Increasing the concentration of deuterium in the medium to 500 ppm provoked higher cytotoxicity compared to medium with natural deuterium content (150 ppm) after 72 h of incubation. There was a slowdown of the cell cycle and a decrease in the rate of cell proliferation, associated with a significant decrease in metabolic activity. At the same time, the reduction of deuterium concentration in the medium to 75 ppm resulted in an increase in metabolic activity and increased proliferation after the same 72 h of incubation compared to the cells cultured under natural levels of deuterium in the medium [53]. A similar effect was demonstrated on mouse splenocytes cultured in medium with reduced deuterium content. Splenocytes in DDW medium had a significantly higher growth rate than cells cultured in normal medium [54]. In vivo experiments have revealed that 24 h consumption of DDW with 10 ppm deuterium content significantly enhances thymus output through induction of both T-cell proliferation and differentiation indicative of immediate cell response [55].

One of the explanations for such biological effects is the Grotguss mechanism, which states that an excess proton or proton defect diffuses through the hydrogen bond network of water molecules or other hydrogen-bonded liquids through the formation and concomitant cleavage of covalent bonds involving neighboring molecules [56]. When the hydrogen proton is replaced by deuterium, the transmission and rotation properties will change. The enrichment of cells with deuterium may lead to inhibition of electron transporting in respiratory chains and, consequently, to the accumulation of electrons in mitochondria and migration to the cytosol, resulting in the generation of free radicals. As the deuteron dissociates slower than the proton, protons in the open sites of macromolecules (DNA, RNA, proteins, etc.) can exchange with deuterons. Deuteron has twice the mass of a proton, and it creates stronger and shorter covalent bonds. The resulting deuteron bonds cause changes in the conformation, and consequently, in the functions of macromolecules. For example, deuterating the active centers of enzymes can provoke changes in their functional activity. Thus, metabolic processes in the body can be changed (stimulated or inhibited) by both decreasing and increasing the content of heavy isotopes compared to their natural levels in the environment [57]. In addition, research has shown that the incubation of cells in water with the shifted ratio D/H, changes not only a parity of deuterium and protium water inside the biological system as the solvent but also isotope exchange in hydroxyl, sulfhydryl, carboxyl, and amino groups of molecules of all organic compounds, which results in the altered rates of biochemical processes [58,59]. At the same time, changes in cell activity can be explained by the replacement of deuterium by protium (or protium by deuterium) in the active and allosteric centers of enzymes, as well as by changes in the deuterium and protium concentrations in the hydrate shell of proteins and nucleic acids, which can change their thermodynamic and, therefore, thermokinetic parameters and thereby change the rate of metabolic and mitogenic processes in cells.

When considering the effect of exposure to water with increased deuterium content on normal cells in the body, the concentration of deuterium is primarily important. The first experiments with 99% DEW on cultures of liver cells and fibroblasts of chicken embryos revealed the death of cultures at 3–4 days [60]. However, the use of the lower concentration of 350 ppm in the later studies on the explants of chicken embryos led mainly to the stimulation of cell growth. This was particularly pronounced for hepatocytes and myocytes [61]. The growth stimulation of these cells can be attributed to the fact that they contain a large number of mitochondria, which react markedly to shifts in peroxidation processes as well as the isotopic D/H gradient, activating autonomous mechanisms of prooxidant–antioxidant state regulation for adaption to new conditions [62].

## 6. Effect of D/H Shift on Proliferation and Apoptosis of Tumor Cells

In addition to the activating effects of isotopically altered water described in various studies on healthy cells and body tissues, a number of scientific papers have investigated the effects of DDW and DEW in neoplastic and metabolic disorders.

A large proportion of in vitro, in vivo, and clinical studies have demonstrated the antitumor effects of the deuterium depletion of cells. A recent study showed that reducing the concentration of deuterium in the cellular environment to 30–100 ppm can lead to regression of a mammary tumor [25]. Reduced deuterium water has been shown to affect the cell cycle, decreasing the number of cells in the S phase and significantly increasing the cell population in the G1 phase, as well as inducing antioxidant enzymes. A similar effect was previously demonstrated on nasopharyngeal carcinoma cells in vitro with a deuterium concentration of 50–100 ppm [63]. DDW significantly inhibited proliferation and reduced the colony-forming invasion capacities of the cells. There was also a cell cycle arrest with a decrease in the number of cells in the S phase and a significant increase in the number of cells in the G1 phase. It is particularly interesting that instead of growth inhibition, DDW promoted the growth of normal control cells in the same experiment. In another study, however, no significant inhibitory effect of DDW with concentrations from 40 to 128 ppm on breast, prostate, gastric cancers, and glioblastoma cell lines was observed [64]. The effect was probably not achieved due to short-term exposure (less than 24 h). A study of Ehrlich ascites tumor-bearing mice showed a significant recovery of metabolic processes fifteen days after injections of tumor cells into the liver in animals that consumed DDW. Liver lipid peroxidation, sialic acid and protein carbonyl levels, xanthine oxidase, myeloperoxidase, catalase, gamma-glutamyl transferase, sorbitol dehydrogenase, glutathione peroxidase and glutathione reductase, glutathione levels and paraoxonase1, sodium potassium ATPase, glutathione-S-transferase, alanine transaminase, and aspartate transaminase were studied. Compared with the tumor group consuming tap water, the DDW-consumed group reversed changes in all parameters except sialic acid, catalase, alanine transaminase, and aspartate transaminase [65].

Thus, the reduction of deuterium concentration in the internal environment is able to regulate the cell cycle of both normal and tumor cells. It is known that the transport of hydrogen ions across the plasma membrane involves H+-ATPase (adenosine triphosphatase), which cannot provide transition of deuterium atoms with the same rate as protium ones [66]. It is likely that when a cell eliminates H+ ions by activating Na+-H+ transport [67,68], the D/H-ratio in intercellular environment increases, thus slowing down the molecular mechanisms, leading the cell to the S phase. However, this fact does not explain the difference in the growth rate of the tumor and normal cells when the deuterium concentration in the medium is reduced. Later investigations have showed that the incubation of malignant cells in DDW for 48 h results in disbalance between reactive oxygen species production and neutralization in mitochondria, thus, leading to oxidative stress in the cells, which activates apoptosis, but the effect was pronounced in some and not all studied malignant cell lines [69].

A possible mechanism of DDW interference in the cell cycle may also underlie the downregulation of expression of certain genes. Some researchers attribute the suppression of tumor cell proliferation to the direct effect of DDW on the level of activation of pro-apoptotic and anti-apoptotic proteins. For example, several studies have clearly demonstrated a decrease in gene expression of Bcl2, Kras, and Myc proteins in various tumor cells. These proteins are among the most important regulators of tumor survival, and significant suppression of their expression leads to a decrease in proliferative activity and increased apoptosis of cancer cells [24,70]. A recent investigation of the expression of 236 cancer-related and 536 kinase genes allows the authors to suggest that the likely mechanism in which the consumption of DDW keeps the D-concentration below natural levels is by preventing the D/H ratio from increasing to the threshold required for cell division [71].

The effects of reduced deuterium levels in the organism on mitochondrial function have also been showed in some studies. The transformation of cells into malignant cells is also associated with the development of mitochondrial defects that result in a lack of hydration of the tricarboxylic acid cycle intermediates with low deuterium matrix water. All oncometabolites not related to the tricarboxylic acid cycle arise from aerobic glycolysis, glutaminolysis, or single-carbon metabolic cycles [72,73,74,75]. Deuterium depletion during metabolic water production is one of the important functions of the mitochondrial matrix. It has been hypothesized that in tumor cells, the balance between the activated H+-transport system and the DDW-producing mitochondria, which determine the D/H ratio of the cells, is disturbed. This disruption leads to accumulation of deuterium in the cell and causes aneuploidy, formation of undifferentiated blast cells, and changes in the size and function of nuclear DNA. Reducing the concentration of deuterium in the cell due to the consumption of DDW restores the disturbed ratio D/H and deeply changes the cellular phenotype and proliferation [76,77].

Several in vivo studies have also confirmed the anti-cancer effect associated with lowering deuterium levels in the internal environment. For example, in a double-blind, randomized 4 month clinical trial of stage II cancers of different locations, the administration of DDW caused a three–seven-fold increase in the mean survival time in lung cancer and a two-fold increase in breast cancer and effectively prevented early stage breast cancer recurrence [78]. In addition, previous clinical studies on patients with various types of tumors showed that consumption of 10–20 ppm DDW caused the arrest of malignant cell growth and significantly increased patient survival, as well as improving quality of life [79]. In this regard, some authors suggest that water with reduced deuterium concentration can selectively affect tumor cells due to its ability to regulate the activity of key genes in the cell cycle, without, however, interfering with metabolic processes in normal tissues [79]. In addition to the expression of genes that regulate cell cycle, cell cycle arrest during transition from the G1 to S phase can be explained by changes in the activity of antioxidant protection enzymes (catalase, superoxide dismutase), the content of which in tumor cells increases with the decrease in deuterium levels [25].

DEW also leads to the increased apoptosis of malignant cells. A study of DEW at concentrations ranging from 10,000 to 50,000 ppm on murine astrocytoma cells induced by Raus sarcoma virus has revealed its pronounced cytotoxicity. The mechanism of D_2_O-mediated cytotoxicity involved the induction of apoptosis and accumulation of the cells in the G2/M phase. The evaluation of rat basophilic leukemia growth in deuterated medium also revealed a lower rate of cell proliferation, which was attributed to the impaired doubling of DNA content [79]. It has been observed that D_2_O-induced apoptosis is modulated through the caspase activation pathway, and its rate is directly proportional to DEW concentration [80]. Deuterium-induced stress also disturbs energy metabolism. A number of studies have confirmed that deuterium elevation affects enzymes involved in energy metabolism, such as cytochrome c oxidase in the mitochondrial respiratory chain [81], as well as on ATP synthase, which leads to a marked decrease in cellular ATP reserves [82]. Primarily, this effect is critical for tumor cells.

Studies of the molecular mechanisms of metabolism and proliferation in deuterated cells show that along with metabolic processes, the structure of organelles gradually changes. After prolonged incubation in D_2_O, cells of *Schizosaccharomyces pombe*, a species of yeast, exhibited rough morphological changes such as thickening of cell walls and aberrant organization of the cytoskeleton [83]. Using transcriptomics and genetic screening, solvent replacement was shown to activate two signaling pathways: the heat–shock response pathway and the cell integrity pathway. In the analysis of differentially expressed genes, a strong D_2_O-dependent repression of some cytosolic and mitochondrial transport RNAs was observed, with the pattern of gene expression triggered by D_2_O resembling that observed in heat shock. However, prior activation of this stress response pathway by heating the cell culture does not confer any advantage in cellular adaptation under solvent replacement conditions. Moreover, instead of protecting the cells against D_2_O, there is a slight decrease in D_2_O tolerance, indicating that the induction of molecular chaperones, oxidoreductases, and other proteins that mitigate the negative effects of stress exposure cannot counteract the adverse effects of D_2_O. However, the restriction of D_2_O-upregulated triggers which activate the cell integrity pathway allows cells to grow, even when H_2_O is completely replaced by D_2_O. This signaling pathway directly controls enzymes participating in the biosynthesis of membrane components, regulates outer cytoplasmic membrane-related gene transcription, and the polarization of the actin cytoskeleton [84,85,86,87].

## 7. Discussion

Thus, a review of publications shows that there are several possible mechanisms of influence of deuterium content on cell proliferation and apoptosis. Some of them seem to be temporary and can be leveled, such as oxidative stress. The maintenance of some others requires further activation of gene transcription, as is the case with an increase in the intensity of aerobic glycolysis in neoplastic cells. A synthesis of the data allows some assumptions to be made about the mechanisms by which deuterium affects cells. The mode and degree of changes in cells and tissues with both an increase and a decrease in the concentration of deuterium in the internal environment depends primarily on the time of exposure, as well as on the concentration. At the initial stage of exposure to both DDW and DEW leads to a change in the D/H gradient between the biological fluids and the intracellular environment, leading to the activation genes regulating metabolic processes and cell cycle progression (Figure 2). This explains the stimulating effect on the organism of water with increased and decreased levels of deuterium in the first few days. Rapid changes in cell proliferation and metabolism indicate that that the D/H gradient is evolutionary old and therefore a well-recognizable parameter by cells. Among diverse cell types, the immune system cells demonstrate higher sensitivity to the first changes in deuterium concentration. Malignantly transformed cells also perceive changes in the D/H gradient but react in another way because initially, anabolic and catabolic processes in neoplastic cells proceed differently with the predominance of glycolysis over pyruvate decarboxylation and the tricarboxylic acid cycle for rapid energy production, necessary for increased proliferation. Apoptosis seems to be mainly involved indirectly by changes induced in metabolism and proliferation. This hypothesis allows us to explain the similarities and differences in effects observed in living organisms after shifts in the D/H gradient and pronounced changes in the deuterium body content, implicating the substitution of hydrogen isotopes in biopolymers.

Further changes in the concentration of deuterium in organisms leads to changes in its content in the intracellular environment and then in the active enzyme centers and in the hydrogen bonds between the nitrogenous base pairs of the DNA molecule. Apparently, deuterium plays a role in controlling the construction of cell membranes. It also may result in a decrease in the proliferative activity of tumor cells in vitro, as well as the maintenance of deuterium levels in tissues while reducing its plasma. Thus, a further increase in deuterium levels leads to a decrease in the proliferation rate and suppression of metabolic processes in various tissues (Figure 2). Such rearrangements of the biochemical processes in tissues with high metabolic activity can affect the functional parameters of an organism. That is why in vivo investigations of organ physiology during gradual and sharp alterations in deuterium content in an organism are essential. Not yet identified and already known effects of deuterium reduction/enrichment may evoke both sanogenetic and pathological processes, for example, the activation of latent diseases. Nevertheless, there is no doubt that a change in the deuterium body content is an element of a complex multilevel system of cell proliferation and apoptosis regulation, which might later be used in therapy.

## 8. Conclusions

The reviewed data show that plant and animal cells are sensitive to deuterium content. Any shifts in the D/H balance outside or inside cells promote immediate responses. The altered rate of proliferation and apoptosis indicate the pivotal role of hydrogen isotope content for normal living and suggest the presence of D/H sensors, which are yet to be detected.

## Figures and Tables

**Figure 1 ijms-24-03107-f001:**
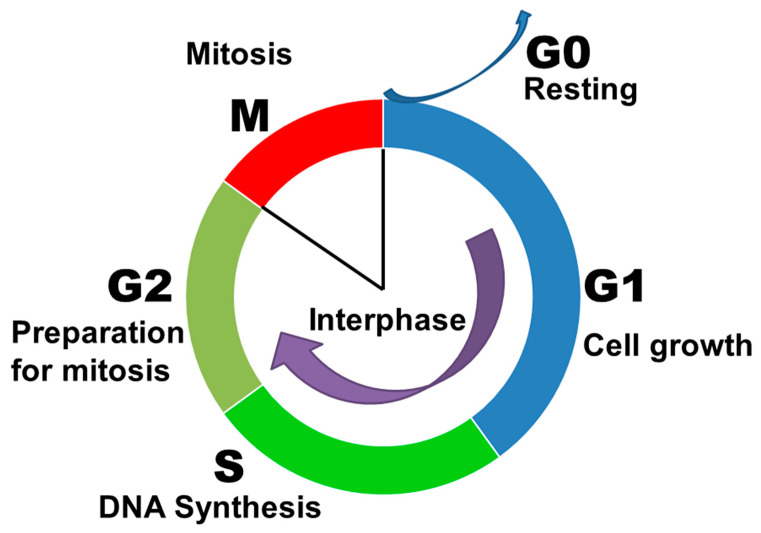
Cell cycle of eukaryotic cells.

**Figure 2 ijms-24-03107-f002:**
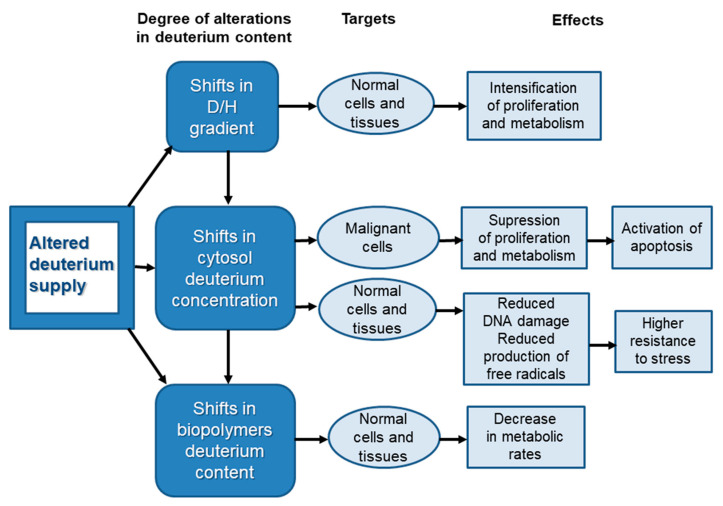
Probable mechanisms of changes in cytophysiology by shifts in deuterium content.

## Data Availability

The data presented in this study are available from the corresponding author upon reasonable request.

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
