# Peer review of "Emerging Role of Deuterium/Protium Disbalance in Cell Cycle and Apoptosis"

_ijms, 2023, doi:10.3390/ijms24043107_

Round 1

Reviewer 1 Report

Yaglova et al wrote a review, outlining the role of deuterium/protium disbalance in cell cycle and apoptosis. However, the focus seem to be more on deuterium content and effect of D/H shift. In addition, it would be useful to further elaborate of the emerging role and its clinical implications [i.e on Section 5 and 6]

Was a systematic search conducted for this review or was it based on expert opinion? Suggest to include how the search was conducted in the earlier sections (what search terms was used, publication from which year) or some details how review was conducted (if applicable)

Were there different measurement of D/H and DDW, DEW across the studies? It would be good to elaborate if the variance observed could be due to different measurements, as well as ways to harmonise the quantitative methods as this may influence the direction and magnitude of change/association observed

An interesting application was reducing the concentration of deuterium to 30-100ppm for anti-tumor activity but readers may question the clinical feasibility of this [Page 7 Line 282]and the eventual implementation

Strongly suggest to include a main Table to summarise the findings from literature: what are the reported studies on effect on D/H on cell proliferation vs apoptosis; in normal vs tumor cells; in vivo vs in vitro; in plants vs animal cells; mouse vs human. This will provide an important comparison and reference*

Some recent reviews such as Zlatska et al 2020 provided a review of effect of deuterium (in vitro) while Zhang et al 2019 provided a review of anticancer effect of DDW leading to oxidative stress, suggest for authors to take reference for how some of the sections may be organised

Suggest to move Fig 2 to discussion and to explain at length the proposed probable mechanisms of changes in cytophisology, the thoughts behind the rationale of this thinking and a critical appraisal of the evidence supporting this postulation of the underlying biological mechanisms. Conclusion should serve to provide a concise summary and synthesis of the main findings

Reviewer 2 Report

The authors have summarized the processes and mechanisms which are influced by hydrogen/deuterium. This includes many cellular processes and the authors have searched the literature to find examples of the effect of introducing depleted or enriched water to these systems and drawn conclusions or summaries on the work cited. Sufficient citations exist and the work is well reported. See attached report for suggested corrections.

Author Response

Thank you for your valuable comments on our manuscript. We have corrected all your comments.

Reviewer 3 Report

The review by Yaglova and coworkers describes the effects of deuterium/protium disbalance on cell cycle and apoptosis. The manuscript is well-organized and well-written. Since quantum mechanical behavior in biology, such as tunneling in hydrogen-transfer reactions, has attracted much attention in recent years, the topic of this review might be of broad interest of readers of International Journal of Molecular SciencesThere are format errors that need to be addressed by the authors or may be corrected by the editorial office before publication in the journal. For example, “14N,” “H2O,” “D2O,” and “Na+” should be “14N,” “H2O,” “D2O,” and “Na+,” respectively.

Minor points:

1. Line 218: “Arbacia punctulate” would be in italic.

2. Line 288: Please remove the second “and” to be “capacities of the cells.”

Author Response

(The authors gave the same response as above.)
